# An efficient and multiple target transgenic RNAi technique with low toxicity in *Drosophila*

Huan-Huan Qiao[1,2], Fang Wang[1], Rong-Gang Xu[1], Jin Sun[1,2], Ruibao Zhu[1,2], Decai Mao[1,3], Xingjie Ren[4], Xia Wang[1], Yu Jia[1,2,7], Ping Peng[1,2,7], Da Shen[1], Lu-Ping Liu[1,5], Zhijie Chang[6], Guirong Wang[7], Shao Li[8], Jun-Yuan Ji[9], Qingfei Liu[10] & Jian-Quan Ni[1,11]

Being relatively simple and practical, *Drosophila* transgenic RNAi is the technique of top priority choice to quickly study genes with pleiotropic functions. However, drawbacks have emerged over time, such as high level of false positive and negative results. To overcome these shortcomings and increase efficiency, specificity and versatility, we develop a next generation transgenic RNAi system. With this system, the leaky expression of the basal promoter is significantly reduced, as well as the heterozygous ratio of transgenic RNAi flies. In addition, it has been first achieved to precisely and efficiently modulate highly expressed genes. Furthermore, we increase versatility which can simultaneously knock down multiple genes in one step. A case illustration is provided of how this system can be used to study the synthetic developmental effect of histone acetyltransferases. Finally, we have generated a collection of transgenic RNAi lines for those genes that are highly homologous to human disease genes.

[1] Gene Regulatory Lab, School of Medicine, Tsinghua University, 100084 Beijing, China. [2] Tsinghua University-Peking University Joint Center for Life Sciences, 100084 Beijing, China. [3] Sichuan Academy of Grassland Science, 611731 Chengdu, China. [4] Department of Neurology, Institute for Human Genetics, University of California San Francisco, San Francisco, CA 94143, USA. [5] Tsinghua Fly Center, Tsinghua University, 100084 Beijing, China. [6] State Key Laboratory of Membrane Biology, School of Medicine and the School of Life Sciences, Tsinghua University, 100084 Beijing, China. [7] State Key Laboratory for Biology of Plant Diseases and Insect Pests, Institute of Plant Protection, 100193 Beijing, China. [8] MTCM-X Center/Bioinformatics Division, Department of Automation, MOE Key Laboratory of Bioinformatics, BNRIST, Tsinghua University, 100084 Beijing, China. [9] Department of Molecular and Cellular Medicine, College of Medicine, Texas A&M Health Science Center, College Station, TX 77843, USA. [10] School of Pharmaceutical Sciences, Tsinghua University, 100084 Beijing, China. [11] Tsingdao Advanced Research Institute, Tongji University, 266000 Qingdao, China. These authors contributed equally: Huan-Huan Qiao, Fang Wang, Rong-Gang Xu, Jin Sun, Ruibao Zhu, Decai Mao. Correspondence and requests for materials should be addressed to J.-Q.N. (email: nijq@mail.tsinghua.edu.cn)

The continued creation of unique and ingenious genetic tools by many drosophilists over the past century has transformed *Drosophila* into a powerful model organism for diverse topics in biomedical research. In the past decade, we have witnessed the development and optimization of the transgenic RNA interference (RNAi) approach, which uses the Gal4/UAS system to control the expression of RNAi constructs, thereby depleting the transcripts of interest in a tissue- or developmental stage-specific manner[1–5]. Due to its relative simplicity and versatility compared to the classical mutants, several large *Drosophila* transgenic RNAi collections have been generated, and their applications have made a major contribution to our understanding of the functions and regulation of genes in developmental and physiological contexts[1–5]. These applications by many fly laboratories have also revealed the intrinsic limitations of these transgenic RNAi lines, such as false negative results for certain highly expressed genes because of poor efficiency, false positive results due to either off-target effects or the leaky expression of the basal promoter, and the unsolved challenge to simultaneously target multiple genes[1,6–12]. Therefore, there is heightened interest in optimizing this transgenic RNAi approach to further increase its efficiency, specificity, and versatility.

Here, we report the development and characterization of a transgenic RNAi system based on a vector named pNP. This system works both in soma and germline with significantly reduced basal level transcription, thereby avoiding potential developmental defects due to leaky hairpin expression. We compare this system with the transgenic RNAi system based on the pVALIUM20 vector which has been widely used by the fly community[4,12]. Using the same Gal4 lines, we observe that the hairpins expressed in the pVALIUM20 system resulted in no obvious phenotypes or lethality before adulthood, but using pNP system to drive the expression of the same hairpins led to strong and specific tissue phenotypes. Moreover, the pNP transgenic RNAi system can efficiently target genes that are highly expressed and produce expected phenotypes. These observations support the greater efficiency of this pNP system compared to the pVALIUM20-based transgenic RNAi system.

This transgenic RNAi system can simultaneously deplete multiple genes, which allows us to study the functions of redundant genes or protein complexes. Moreover, the loss-of-function phenotypes produced by targeting two genes simultaneously are more robust than the traditional method to express hairpins using one recombinant carrying two UAS cassettes. To provide a case illustration of this system, we focus on the histone acetyltransferases (HATs), since depleting individual HATs in the *Drosophila* eye using RNAi fails to generate any phenotypes. Surprisingly, simultaneously knocking down the transcripts of three HATs, *chameau* (*chm*), *Tip60*, and *Gcn5*, by the pNP system generated phenotypes with severe eye defects. Our subsequent analyses revealed that these HATs work together in modulating Wnt signaling.

Finally, we generate a resource of reagents against the genes that are highly homologous to human disease genes. We anticipate that this transgenic RNAi system and the relevant resource will further aid the developmental genetic analysis of the functions and regulation of *Drosophila* genes in the future.

## Results

### The pNP system works efficiently in soma and germline.
The conditional transgenic RNAi systems in *Drosophila* can temporally and spatially knock down target genes with the binary Gal4/UAS system. Without Gal4, the hairpin encoded by the DNA sequence downstream of the UAS would not be expressed, enabling us to generate and maintain thousands of transgenic

RNAi lines based on the pVALIUM vectors[2,4]. However, we noticed that 15.8% of the transgenic RNAi lines based on the pVALIUM20 system in the Tsinghua Fly center (THFC) cannot maintain homozygosity (Supplementary Data 1). Meanwhile, 8.5% of the φC31 and long double-stranded RNA (dsRNA) hairpin based transgenic RNAi library (KK collection) in the Vienna Drosophila Resource Center (VDRC) are heterozygous, and 17.9% of the pVALIUM20 system based transgenic RNAi lines in the VDRC are heterozygous (Supplementary Data 1). All these vectors used in the previously developed transgenic RNAi systems are based on the pUAST vector, whose promoter consists of tandem UAS repeats followed by the TATA box from the *Drosophila hsp70* gene[1,4]. Given that the *hsp70* basal promoter can direct transcription without Gal4 induction in certain tissues at the standard 25℃[9], we speculated that the lethality of these homozygous transgenic RNAi lines could be due to the leaky activation of the *hsp70* basal promoter.

To test this possibility, we replaced the *hsp70* basal promoter in the pVALIUM20 vector with a *Drosophila* synthesized core promoter (DSCP)[13]. Since the *hsp70* basal promoter is longer than DSCP, we added a 100 bp spacer sequence upstream of the UAS to ensure the appropriate working distance between the promoter and gypsy insulators. To achieve precise processing and expression of multiple shRNAs[14], we amplified an intergenic linker between miR-2a-1 and miR-2b-2 and cloned it downstream of the miR1 scaffold. With these optimizations, we designated the resulting construct as the pNP vector (Supplementary Fig. 1).

We designed and generated stable transgenic lines using this system to express shRNAs targeting a set of protein-coding genes with known loss-of-function phenotypes in the wing, eye, neurons, intestine, testis and ovarian stem cell system. Firstly, knockdown of *egfr, Hp1a, kis, Notch* in the wing driven by wing specific Gal4s generated severe wing defect (Fig. 1a), which is consistent with previous description[4,15–18]. Furthermore, targeting of *white, light, hedgehog* (*hh*) using the ey-Gal4 produced the same eye phenotype as previous reports (Fig. 1b)[19–21], suggesting the pNP transgenic RNAi system can efficiently deplete genes in soma. We also chose several genes with known loss-of-function phenotypes to analyze the efficiency of pNP system in the neurons. Consistent with previous reports[1,22,23], all of them caused lethality when driven by elav-Gal4, indicating that pNP works efficiently in neurons (Supplementary Fig. 2a). Next, we tested the efficiency of the pNP system in intestine, testis and ovarian stem cell system. We used the germ cell specific nos-Gal4 to knock down *piwi* or *bam*, which severely reduced ovary size compared with the control (Fig. 1c), and resulted in flies that did not lay eggs (Fig. 1d). In addition, RNAi of *bam* increased the number of germline stem cells (GSCs) as previous reports[24], while knocking down of *punt* in testis by nos-Gal4 showed no GSCs and less germ cells[25], and depletion of *Notch* in intestine by esg-Gal4 increased the number of intestinal stem cells and enteroendocrine cells (Supplementary Fig. 2)[26]. Taken together, these results show that the pNP transgenic RNAi system can efficiently and specifically knock down target genes in not only soma, but also neurons, intestine, testis and ovarian stem cell system.

### Significantly reduced leaky expression from basal promoter.
To compare the basal activities of the promoters in the pNP and pVALIUM20 vectors, we replaced the miR1 scaffold with the *luciferase* gene as the reporter. First, we tested the leaky expression of luciferase in these transgenic flies at the different developmental stages without the Gal4 driver. As shown in Fig. 2a, the luciferase intensities from the pNP-*luciferase* transgenic flies are

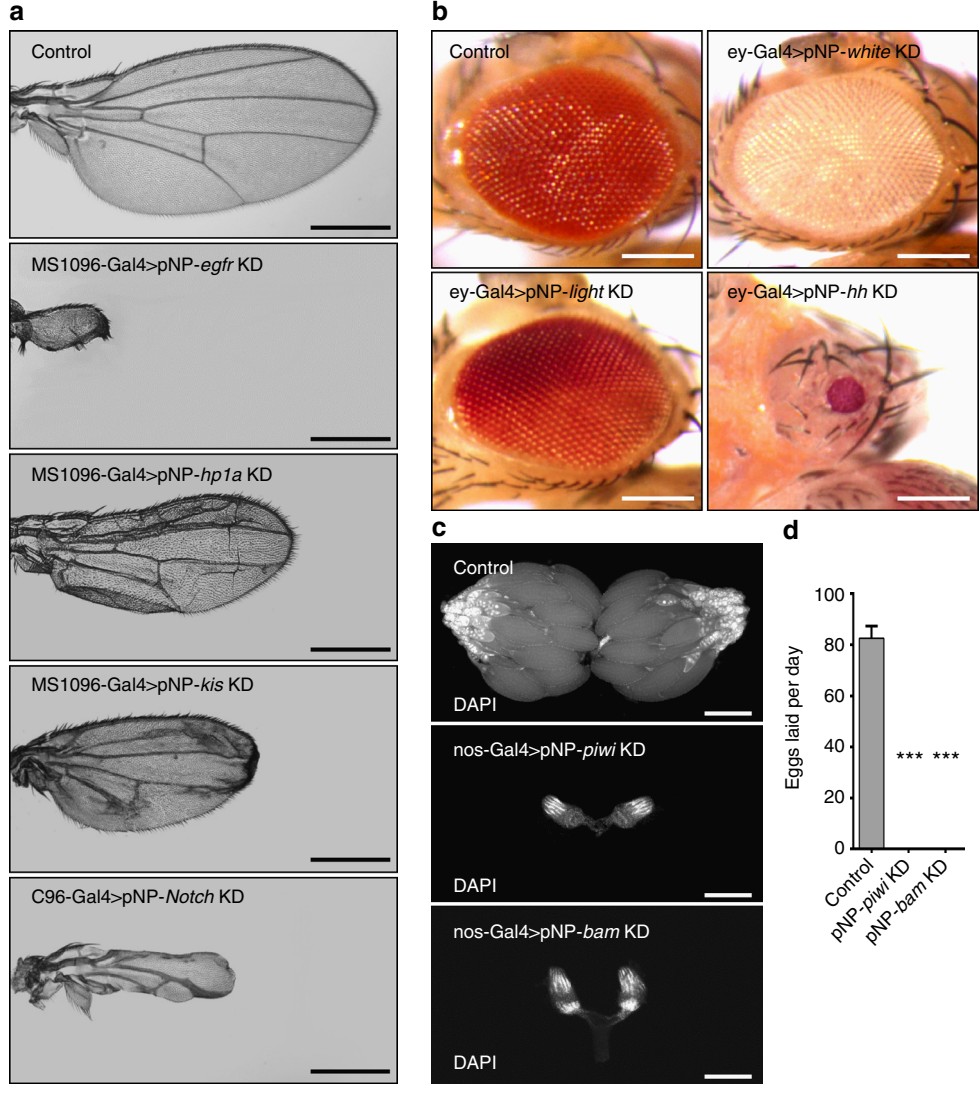

**Fig. 1** The pNP system works efficiently in soma and germline. **a** Examples of pNP-induced wing phenotypes using the MS1096-Gal4 or C96-Gal4. Scale bars, 500 μm. **b** Knock down of *white, light* or *hh* in the eye driven by ey-Gal4. Scale bars, 200 μm. **c** Dark field images of ovary phenotype, with knock down of *piwi* and *bam* controlled by nos-Gal4. Scale bars, 300 μm. **d** Fertility rates of control, pNP-*piwi* and pNP-*bam* female flies using the nos-Gal4 driver (*n* = 10, mean ± s.d.). Data are evaluated with one-tailed Student's *t*-test (*$p < 0.05$, **$p < 0.01$, ***$p < 0.001$)

similar to those of the wild-type animals at all developmental stages analyzed, but their levels are all dramatically increased in pVALIUM20-*luciferase* animals after the embryonic stage. To test the potential tissue-specific differences, we dissected different tissues from the third instar larvae and performed the luciferase assays. Similar luciferase intensities were observed in the muscle and the central nervous system (brain and nerve cord) with both vectors (Fig. 2b). However, we observed the significantly elevated basal expression of luciferase in salivary gland, male gonad, female gonad, fat body, and wing disc from the pVALIUM20 system, being 685, 41, 42, 167, and 4 fold higher than in the corresponding tissues from the pNP system (Fig. 2b).

To validate these observations based on the luciferase reporter, we compared the levels of heterochromatin protein 1 (HP1) by immunostaining using the polytene chromosomes from the pVALIUM20-*Hp1a* and pNP-*Hp1a* larvae without Gal4. Compared with the control and pNP-*Hp1a*, the levels of HP1a on polytene chromosomes from the pVALIUM20-*Hp1a* larvae were significantly decreased (Fig. 2c), and the relative HP1a intensity at the chromocenter was down to 20% (Fig. 2d), suggesting the leaky expression of shRNA in pVALIUM20-*Hp1a*. Consistent with this

observation, we used the qRT-PCR assay and observed that the *Hp1a* mRNA levels in the salivary gland of pVALIUM20-*Hp1a* larvae were also significantly reduced compared to the control, while no differences in *Hp1a* mRNA levels were observed between pNP-*Hp1a* larvae and the control larvae at the same developmental stage (Fig. 2e). Taken together, these data support that, compared with the pVALIUM20 vector, the pNP system had significantly reduced the leaky expression.

To further test whether the leaky expression from the pVALIUM20 vector is deleterious to fly development thereby increasing the heterozygous rate of the transgenic animals, we built pVALIUM20 and pNP based transgenic lines against more than 300 genes, respectively (Supplementary Data 2). For each target gene, same shRNA was cloned into pVALIUM20 or pNP vector, respectively, and they were inserted in the same chromosome locus. When comparing the heterozygous rate of these transgenic lines, we observed that almost 15% of the pVALIUM20 system based transgenic lines were heterozygous, which is consistent with the rate in existing stock centers, while the pNP system based transgenic lines exhibited only 0.6% heterozygosity (Fig. 2f). These results demonstrate that the leaky

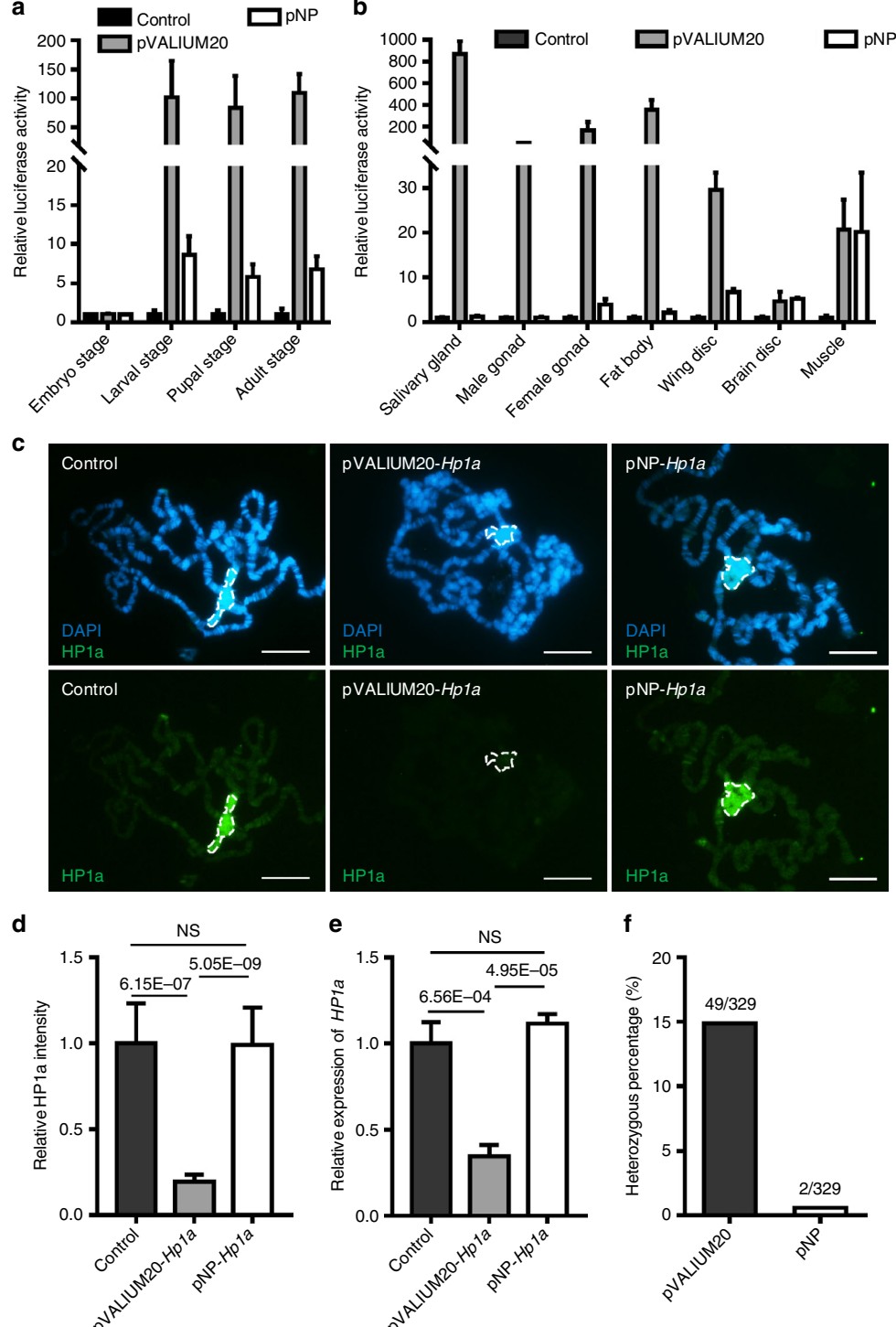

**Fig. 2** Significantly reduced leaky expression from basal promoter. **a** Relative luciferase expression levels at different developmental stages of pVALIUM20-*luciferase* and pNP-*luciferase* flies without the Gal4 driver ($n = 5$, mean ± s.d.). **b** Relative luciferase expression levels in different tissues of pVALIUM20-*luciferase* and pNP-*luciferase* flies at larval stage without the Gal4 driver ($n = 3$, mean ± s.d.). **c** Immunostaining of polytene chromosome from pVALIUM20-*Hp1a* and pNP-*Hp1a* without the induction of Gal4. DNA was visualized with DAPI and HP1a was stained with mouse monoclonal anti-HP1a (HP1a C1A9). Dotted circles indicate chromocenter. Scale bars, 20 μm. **d** Quantification of the relative HP1a intensity at chromocenter of polytene chromosome in pVALIUM20-*Hp1a* or pNP-*Hp1a* without the Gal4 driver ($n = 10$, mean ± s.d.). Data are evaluated with one-tailed Student's *t*-test. **e** Relative expression of HP1a in control, pVALIUM20-*Hp1a* or pNP-*Hp1a* salivary gland without the driving of Gal4 ($n = 3$, mean ± s.d.). Data are evaluated using one-tailed Student's *t*-test. **f** The heterozygous percentage of pVALIUM20 system based transgenic lines and pNP system based transgenic lines

expression is significantly reduced in the pNP system, also decreases the side effect shown by the pVALIUM20 system.

**The pNP system is more efficient than the pVALIUM20 system**. The pVALIUM20 system works well in somatic cells and female germline for many genes; however, shRNAs targeting some of the functional genes in development generated no obvious or only weak phenotypes. To validate whether the pNP system is more efficient than the pVALIUM20 system, first we tested the expression of luciferase in different tissues of pNP-*luciferase* and pVALIUM20-*luciferase* transgenic flies using different Gal4 drivers, including elav-Gal4, MS1096-Gal4, Nub-Gal4, MTD-Gal4, nos-Gal4, GMR-Gal4, and esg-Gal4. As shown in Supplementary Fig. 3, both pNP-*luciferase* and pVALIUM20-*luciferase* transgenic flies showed highly expressed luciferase comparing with control. Furthermore, the luciferase intensities of pNP-*luciferase* transgenic flies are all significantly higher than pVALIUM20-*luciferase* animals using these drivers, indicating the pNP system is more efficient. Then we chose the same shRNAs, targeting genes encoding proteins known for their critical functions during the development of eye, wing or ovary, and generated transgenic lines using the pNP system. The genes targeted were *E2F1*, *Upf1*, *aTub67C*, and *egg*.

In our first example, we tested E2F1, the key transcription factor involved in regulating the G1-S phase transition of the cell cycle[27,28]. Depletion of *E2F1* in the eye using ey-Gal4 in the pVALIUM20-*E2F1* system slightly reduced eye size; in contrast, the pNP-*E2F1* system combined with ey-Gal4 generated a much smaller eye (Fig. 3a). In our second case study, we analyzed Upf1, an important protein involved in nonsense mRNA mediated decay[29], since knocking down *Upf1* using MS1096-Gal4 and the pVALIUM20-*Upf1* system did not generate obvious phenotypes. However, we found that depletion of *Upf1* using pNP-*Upf1* driven by MS1096-Gal4 led to small wings with disrupted anterior cross vein (a-cv) (Fig. 3b). Similarly, depleting *aTub67C* in the wing using the pVALIUM20-*aTub67C* system and Nub-Gal4 failed to generate any phenotypes, presumably due to the high abundance of this transcript in cells. Nevertheless, a similar experiment using the pNP-*aTub67C* system led to extremely small wings (Fig. 3c). These results suggest that the pNP system produces stronger phenotypes than pVALIUM20 in somatic tissues.

To compare the knockdown efficiency of these two systems in the female germline, we crossed the nos-Gal4 line with pNP-*egg* and pVALIUM20-*egg*. As shown in Fig. 3d, the ovaries from pVALIUM20-*egg* females were smaller than controls, but the ovaries from the pNP-*egg* females were further reduced in size (Fig. 3d). Consequently, compared with a few eggs produced from the pVALIUM20-*egg* females, we did not obtain any eggs from the pNP-*egg* females (Fig. 3e). These results suggest that the pNP system also works more efficiently than the pVALIUM20 system in the female germline. Moreover, to exclude the possibility of off-target effect and rescue the phenotype generated in Fig. 3, we overexpressed the target genes that are insensitive to the hairpins in the RNAi flies driven by specific Gal4 lines. As shown in Supplementary Fig. 4, both the pVALIUM20-induced phenotype and the pNP-induced phenotypes were all significantly rescued, respectively, compared with the RNAi phenotypes alone, suggesting the specificity of these RNAi. Taken together, these results support the efficiency and specificity of this pNP system.

To further validate the notion that the pNP system is more efficient in depleting genes of interest than the pVALIUM20 system, we quantified the mRNA levels using the qRT-PCR assay. Although the pVALIUM20 system significantly reduced the transcripts of *E2F1*, *Upf1*, and *egg* compared to the control, greater depletion of these mRNAs was achieved sing the

pNP system (Fig. 3f). For the *aTub67C* gene, the pVALIUM20-system did not affect mRNA level, but significant reduction was observed in wing discs using the pNP system (Fig. 3f). These observations are consistent with the aforementioned phenotypic assays. Remarkably, as a member of the tubulin family, aTub67C is highly expressed in cells; the pVALIUM20 system failed to knock down *aTub67C* transcripts (Fig. 3f), implying the weakness of the pVALIUM20 system in depleting highly expressed genes. Taken together, these results suggest that the pNP system is more efficient than the pVALIUM20 system in both soma and female germline.

**The pNP system efficiently modulates high expression genes**. To efficiently deplete genes that are highly expressed in cells has been a challenge in the field. The observations we made with the *aTub67C* gene prompted us to test additional cases to compare the pNP system with the pVALIUM20 system. For this, we chose a set of multiple-copy genes, *histone H1*, *H2A*, *H2B*, *H3*, and *H4*, all highly expressed in cells. Using the same hairpins for each histone gene, we constructed transgenic RNAi flies based on the pNP system or the pVALIUM20 system. When driven by MS1096-Gal4, knock down of all the histones using the pNP system generated severe wing defects (Supplementary Fig. 5), consistent with the critical role of histones in cell viability. Surprisingly, depleting *H1* or *H3* using the pVALIUM20 system generated weak or no aberrant phenotypes, while knocking down *H2A*, *H2B*, or *H4* in this system caused lethality before the pupal stage (Supplementary Fig. 5). Given that the pNP system is more efficient but less leaky than the pVALIUM20 system, we suspected that the leaky expression of pVALIUM20-*H2A*, *H2B*, and *H4* plus the MS1096-Gal4 induced RNAi in early developmental stages may cause the lethality of these pVALIUM20 transgenic lines.

To test this idea, we selected the Nub-Gal4; tub-Gal80[ts] system to compare the wing phenotypes using pNP and pVALIUM20. Gal80[ts] is a temperature-sensitive repressor of Gal4, under the control of the *tubulin* promoter[30,31]. The crossed flies were first maintained under the permissive temperature 18 °C which allows Gal80 to repress Gal4 activities. The third instar larvae were then shifted to 29 °C, which inactivates Gal80 thereby allowing Gal4-dependent expression to occur normally and deplete the histone genes. As shown in Fig. 4a, we observed the severely disrupted wing phenotype for all the histones depleted using the pNP system, similar to the RNAi phenotypes driven by MS1096-Gal4 (Supplementary Fig. 5). In contrast, the pVALIUM20 system showed no obvious phenotype for any of the depleted histones (Fig. 4a). The qRT-PCR result also confirmed that the pNP system was more efficient than the pVALIUM20 system (Fig. 4b), supporting the pNP system could efficiently modulate high expression genes with low false positive results.

The false positive results in pVALIUM20-*H2A*, *H2B* and *H4* flies were likely from the combination of the leaky expression and MS1096-Gal4 induced RNAi in early developmental stage. In our analysis, pVALIUM20-*H2A*, *H2B* and *H4* driven by MS1096-Gal4 were all lethal (Supplementary Fig. 5), but were viable and generated weak wing phenotypes using Nub-Gal4; tub-Gal80[ts], which prevented RNAi in early developmental stage (Fig. 4a). However, these hairpins driven by MS1096-Gal4 in the more efficient pNP system (also with low leaky expression) were viable (Supplementary Fig. 5) and gave the same severe phenotypes as Nub-Gal4; tub-Gal80[ts] generated (Fig. 4a). Taken together, these results suggest that the leaky expression of the pVALIUM20 system plus MS1096-Gal4 induced RNAi in early developmental stage can cause false positive results.

These observations indicated that the pVALIUM20 system may cause lethality because of leaky and non-specific expression,

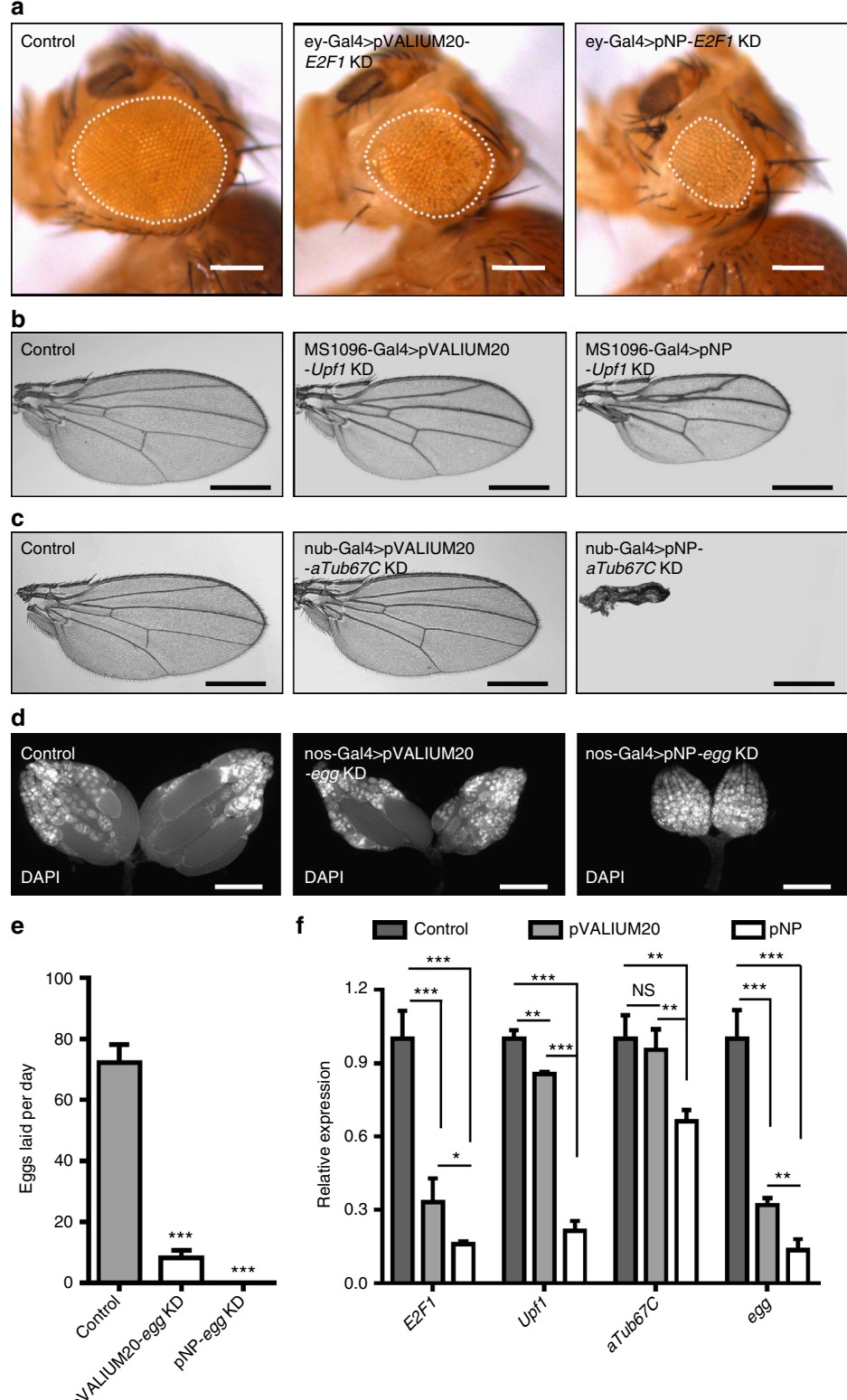

**Fig. 3** The pNP system is superior to the pVALIUM20 system. **a** Knock down of *E2F1* in the eye using the pVALIUM20 system or the pNP system with the same shRNA driven by ey-Gal4. Scale bars, 200 μm. **b** RNAi of *Upf1* in the wing using MS1096-Gal4. Scale bars, 500 μm. **c** Targeting of *aTub67C* with the pVALIUM20 system or the pNP system driven by Nub-Gal4. Scale bars, 500 μm. **d** Knock down of *egg* in the germline controlled by nos-Gal4. Scale bars, 300 μm. **e** Fertility rates of control, pVALIUM20-*egg* and pNP-*egg* female flies using the nos-Gal4 driver ($n = 10$, mean ± s.d.). Data are evaluated with one-tailed Student's *t*-test. **f** qRT-PCR analysis of the RNAi efficiency of *E2F1*, *Upf1*, *aTub67C*, and *egg*, respectively ($n = 3$, mean ± s.d.). Data are evaluated using one-tailed Student's *t*-test (*$p < 0.05$, **$p < 0.01$, ***$p < 0.001$)

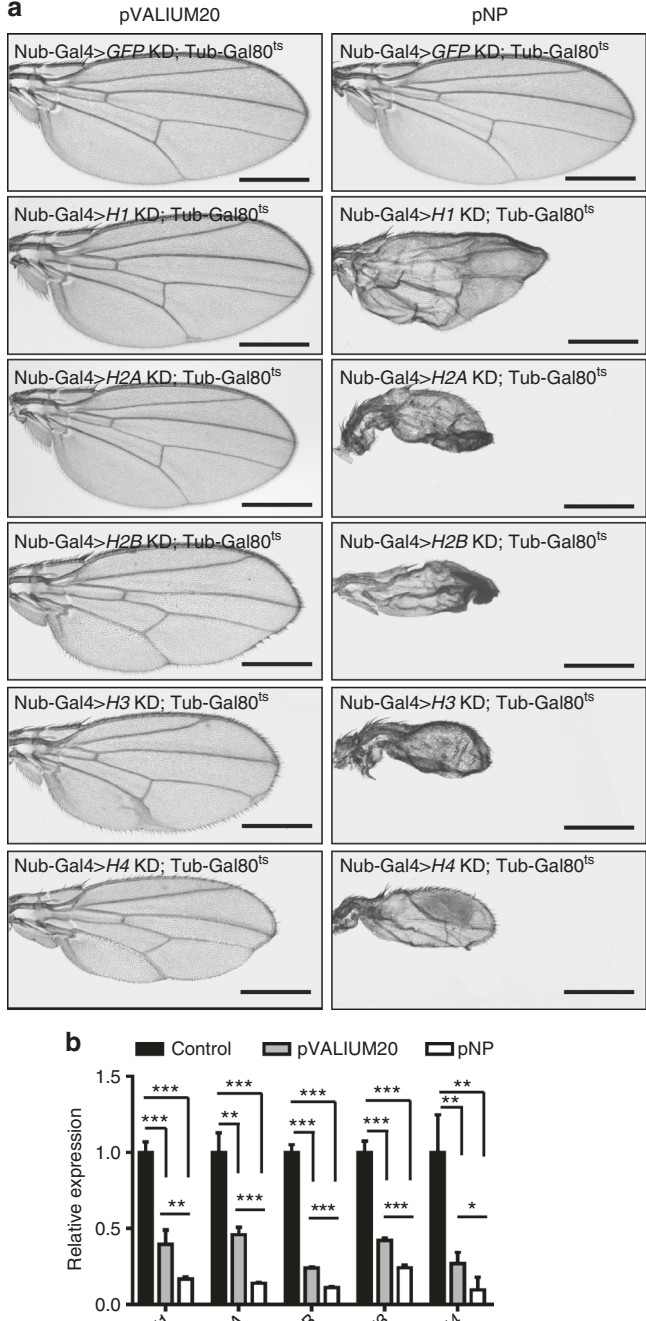

**Fig. 4** The pNP system could efficiently modulate high expression genes. **a** The Nub-Gal4; tub-Gal80ts system was used for conditional knock down of histones in the pNP and pVALIUM20 systems, with the flies first incubated at 18 °C and then shifted to 29 °C at third larval stage. Scale bars, 500 μm. **b** The act-Gal4; tub-Gal80ts system was used for qRT-PCR analysis of the RNAi efficiency, targeting *H1, H2A, H2B, H3,* and *H4* (n = 3, mean ± s.d.). Data are evaluated using one-tailed Student's t-test (*p < 0.05, **p < 0.01, ***p < 0.001)

which can be problematic particularly when applying this system for large-scale genetic screens. For example, we randomly selected 20 genes for which shRNAs in pNP gave wing defect phenotypes if crossed with wing-specific Gal4; three of them were lethal if the same hairpins were used in pVALIUM20, further supporting the high tissue specificity of the pNP system (Supplementary Data 3). Taken together, these results show that the pNP system is more

efficient than the pVALIUM20 system in knocking down genes of interest, including those with high levels of expression, and that it produces the expected strong phenotypes without obvious leaky expression that often observed in the pVALIUM20 system.

**The pNP system can simultaneously target multiple genes.** Proteins encoded by two or more genes in *Drosophila* can have partially overlapping or redundant functions, and thus depleting one of them may result in little or no effect on functional or phenotypic analyses. In addition, depletion of a single subunit of a protein complex may result in the formation of partial protein complexes, which can have dominant-negative or neomorphic effects in vivo. Thus it is advantageous to simultaneously deplete multiple genes. However, existing transgenic RNAi vectors can only accept a single shRNA; therefore, to simultaneously target two or more genes cumbersome genetic crosses are required. Another problem is that genetic recombination may reduce RNAi efficiency because of Gal4 dilution, as well as that unhealthy recombinants frequently occur.

An intergenic linker region between miR-2a-1 and miR-2b-2 was inserted into the pNP vector following the miR1 scaffold (Supplementary Fig. 1), which could express multiple shRNAs. We performed three sets of experiments in parallel, comparing the efficiency to deplete these sets of target genes in the same vector individually or simultaneously. First, we subcloned a hairpin targeting *Notch* gene and a hairpin targeting *white* gene into the pNP vector, designated as the pNP-*N-W* vector, and generated transgenic lines. Same hairpins subcloned into the pNP vector individually were used as positive controls. When driven by the eye-specific GMR-Gal4 line, RNAi depletion of *white* completely eliminated red pigments in the eyes, while knocking down either *Notch* alone or *Notch* and *white* together (*Notch-white*) caused lethality at larval stage (Fig. 5a). If RNAi was induced by the wing-specific C96-Gal4 line, depleting *white* in the wing did not generate any abnormal phenotypes as expected, whereas depletion of either *Notch* alone or *Notch-white* generated the same wing phenotype with reduced size. The observation that the effects on the eye by depleting the *white* gene and the wing effects caused by knocking down *Notch* occur independently within the pNP-*N-W* line demonstrates that the pNP vector can simultaneously target multiple genes with one RNAi vector (Fig. 5b). To test the possibility that the RNAi-induced phenotypes might be affected by the order of the hairpins in the pNP vector, we generated a transgenic line with the *white* hairpin in front of the *Notch* hairpin (pNP-*W-N*), and then compared the RNAi efficiency with that of the pNP-*N-W* line. As shown in Fig. 5a, b, both the pNP-*W-N* and pNP-*N-W* lines generated the identical phenotypes when driven by GMR-Gal4 or C96-Gal4 lines, suggesting that the hairpins in the pNP vector function independently, without a sequential effect.

In addition, we chose another set of protein coding genes, *Ci* and *E2F1*, as depleting both of them using Nub-Gal4 generated distinct phenotypes in the wing. As shown in Fig. 5c, depleting *Ci* using pNP-*Ci* generated anterior fusion between L3 and L4 without affecting the size of the wing, while knocking down *E2F1* using pNP-*E2F1* led to a smaller wing with vein fusion between L2 and L3. When both genes were depleted using either the pNP-*Ci-E2F1* line or the pNP-*E2F1-Ci* line, we observed the same small-sized wing as for pNP-*E2F1*, which was also carrying the vein fusions; one was between L2 and L3 from *E2F1* KD, another was between L3 and L4 from *Ci* KD (Fig. 5c). These phenotypes appear to be due to the additive effect of those caused by depleting either *E2F1* or *Ci* alone, further supporting the high efficiency of the pNP system in targeting multiple genes. Again, no sequential effect was observed by comparing the pNP-*Ci-E2F1*

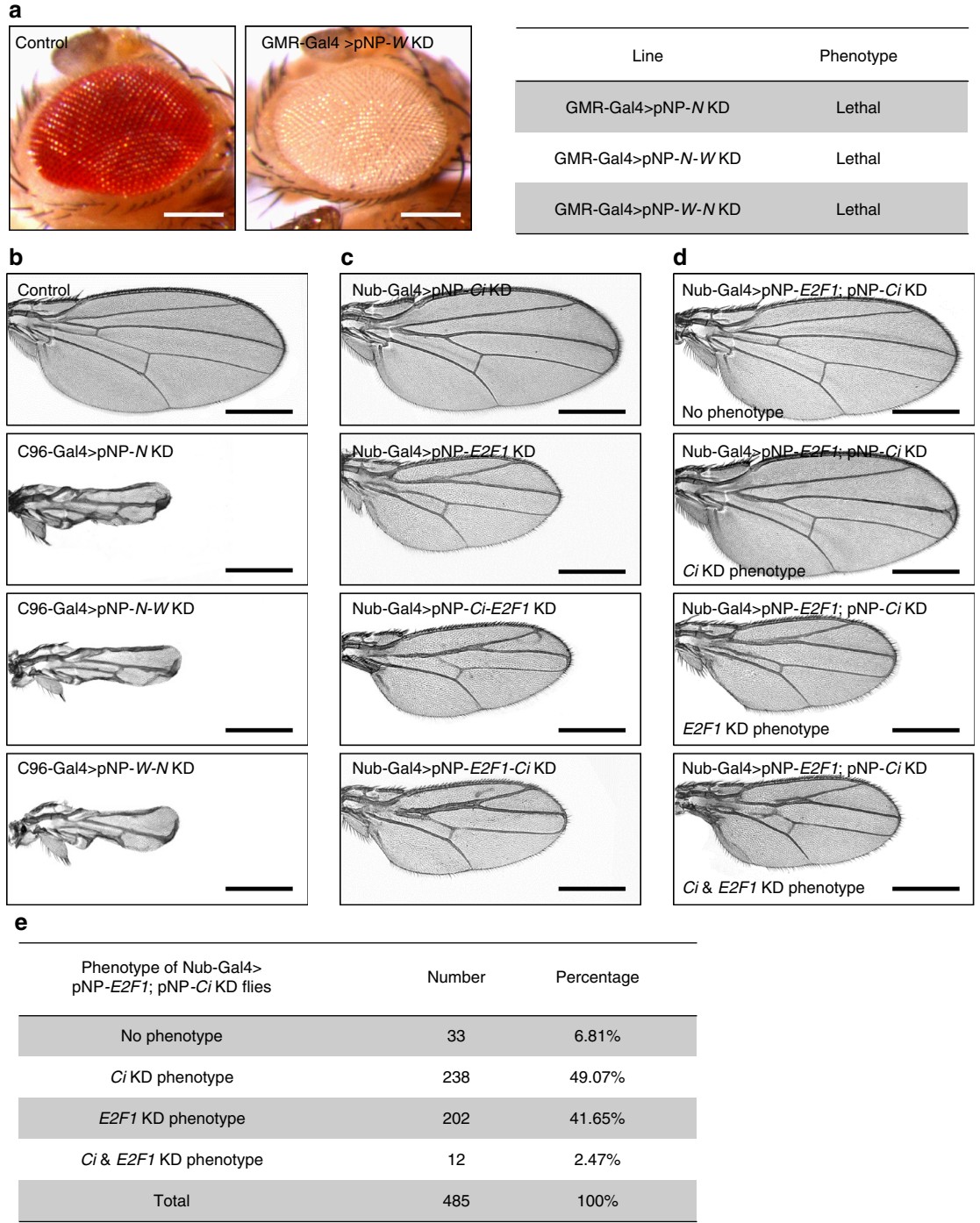

**Fig. 5** The pNP system can simultaneously target multiple genes. **a** RNAi against *white* caused complete loss of red pigment in eye; also note that knock down of *notch* in eye using GMR-Gal4 caused lethality. Scale bars, 200 μm. **b** Compared with control, pNP-*N*, pNP-*W-N* and pNP-*N-W* driven by C96-Gal4 all showed similar *Notch* wing defect phenotypes. Scale bars, 500 μm. **c** Disruption of *Ci* driven by Nub-Gal4 produced anterior fusion of L3 and L4 veins of the wing, while knock down of *E2F1* caused small wing with anterior fusion of L2 and L3 veins. The combined phenotype of anterior fusion of L2, L3, L4 veins and small wing showed in pNP-*Ci-E2F1* and pNP-*E2F1-Ci* flies. Scale bars, 500 μm. **d**, **e** Traditional genetic combination of pNP-*Ci* and pNP-*E2F1* exhibited a much weaker phenotype. Only a small proportion of pNP-*Ci*; pNP-*E2F1* flies showed both *Ci* KD and *E2F1* KD phenotype; most of them generated either *Ci* KD or *E2F1* KD, while there were even some similar with wild-type phenotype. Scale bars, 500 μm

line and the pNP-*E2F1-Ci* line (Fig. 5c). To compare the efficiency of multiple shRNA knock down from the pNP system with traditional genetic recombination, pNP-*Ci* and pNP-*E2F1* were genetically recombined first and then crossed with the Nub-Gal4 line. Interestingly, only 2.47% of the pNP-*Ci*; pNP-*E2F1* flies showed both *Ci* KD and *E2F1* KD phenotypes; most of them generated either the *Ci* KD or *E2F1* KD phenotype alone, while

6.81% in fact produced no obvious phenotype (Fig. 5d, e). This variable result illustrates another drawback of using traditional genetic recombination to knock down multiple genes, presumably due to competition for Gal4 binding by the two UAS cassettes. Moreover, the flies with the integrated line of pNP-*Ci*; pNP-*E2F1* appeared weak as homozygosity was lethal before adulthood, likely due to too many exogenous genetic components. These

observations further support the notion that the pNP system is more robust and versatile than traditional recombination to target multiple genes.

To further validate the ability of pNP in depleting multiple genes simultaneously, we applied this system to Polycomb repressive complex 1 (PRC1), which function as transcriptional repressor complexes essential for normal development[32]. Simultaneous knockdown of four core proteins of PRC1 generated severe eye defects, while depleting them individually did not cause any defects in the eye (Supplementary Fig. 6a). The RNAi efficiency of tetradic-gene-KD is also comparable with single-gene-KD tested by qRT-PCR assay (Supplementary Fig. 6b). Taken together, these results demonstrate the remarkable effectiveness of the pNP system to simultaneously knock down multiple genes.

**HATs regulate eye development through Wnt signaling pathways.** In an effort to test this transgenic RNAi system for studying proteins with partially redundant functions, we chose to study a family of histone acetyltransferases (HATs), the enzymes that mainly acetylate lysine residuals on histones. Given that the HAT family of enzymes regulate transcriptional activation, DNA repair, and the cell cycle, dysregulation of their activities is often linked to epigenetic alterations in many developmental disorders and diseases such as cancer[33]. Transcriptional activation requires multiple HATs, and the functions of some HATs are generally redundant; thus the depletion of one HAT is insufficient to produce phenotypes. We focused our analyses on a set of HATs belonged to the MYST or GNAT families, including *chm, Tip60*, and *Gcn5*. After generating the transgenic RNAi lines against *chm, Tip60, Gcn5*, and the combinations of *chm-Tip60* and *chm-Tip60-Gcn5*, we crossed these lines with the eye-specific GMR-Gal4 line. We observed that the pNP-*chm*, pNP-*Tip60*, pNP-*Gcn5*, and pNP-*chm-Tip60* did not produce any defects in the eye (Fig. 6a). However, simultaneously depleting all three HATs in the eye using the pNP-*chm-Tip60-Gcn5* line led to a severely defective eye phenotype, characterized by the overproduction of eye pigment, necrotic cell death, and slightly reduced eye size (Fig. 6a) This is unlikely to be caused by off-target effects, as depletion of any one of them produced no obvious defects.

To confirm that this phenotype is indeed caused by depletion of *chm, Tip60*, and *Gcn5*, we performed a qRT-PCR assay to measure the transcripts of these genes in eye discs after RNAi triggered by GMR-Gal4. Compared with the control, the transcripts from individual RNAi lines and the triple RNAi line were all significantly reduced (Fig. 6b). Notably, the efficiency of RNAi depletion in the triple RNAi line was comparable to that of the individual RNAi lines, further demonstrating the power of the pNP system to simultaneously target multiple genes. To further exclude the possibility of off-target effects, we overexpressed *chm*-T2A-*Tip60*-T2A-*Gcn5* that insensitive to the hairpins by using the T2A self-cleaving peptide in the *chm-Tip60-Gcn5* triple KD flies, and the KD phenotype was fully rescued (Supplementary Fig. 7b), supporting the specificity of this RNAi experiment. Furthermore, we also generated a transgenic activation line that targeting these three genes simultaneously using the flySAM system we developed recently[34]. As shown in Supplementary Fig. 7c, activation of these three genes also significantly rescued *chm-Tip60-Gcn5* triple KD tumor-like phenotype, further supporting the specificity of this triple KD result.

Furthermore, to validate the enzyme activity of *chm, Tip60, Gcn5* on histone acetylation, we performed immunostaining to detect the level of global H4 acetylation in polytene chromosomes after RNAi induced by salivary gland-specific 1824-Gal4. The intensity of immunostaining against H4 acetylation in polytene chromosomes from the pNP-*chm*, pNP-*Tip60*, pNP-*Gcn5* lines

was slightly lower compared to the control, whereas the levels of H4 acetylation in polytene chromosomes from the pNP-*chm-Tip60-Gcn5* line were almost undetectable (Supplementary Fig. 8). Taken together, these observations demonstrate the redundant roles of these HATs on epigenome, further supporting the ability of the pNP system to successfully modulate multiple genes.

To explore the molecular mechanism of the severe eye defect phenotype produced by triple RNAi against *chm-Tip60-Gcn5*, we detected the transcriptional level of genes downstream of several major signaling pathways, such as the JAK-STAT, EGFR, Hippo, JNK, dpp, Notch, and Wnt signaling pathways, because of their roles in regulating eye development. Specifically, we analyzed the expression of *SOCS36E, aos, Yki, puc, dad, Su(H)* and *wg* as the readouts for the activities of the aforementioned signaling pathways in eye discs with these HATs being depleted[35]. As shown in Fig. 6c, *wg*, a target of Wnt signaling, was significantly up-regulated by more than five-fold of the control, while the representative target genes of the other signaling pathways tested were not significantly affected in HAT-depleted eye discs. These observations indicate that Wnt may be the major signaling pathway being affected by these HATs in eye discs.

To test the effects of depletion of the HATs on Wnt signaling, we generated the *chm-Tip60-Gcn5* KD clones marked with GFP in wing discs and then analyzed the levels of Wg expression using immunostaining. As shown in Fig. 6d, the *chm-Tip60-Gcn5* KD clones showed clearly elevated levels of Wg proteins. Meanwhile, we constructed *wg* transcriptional activation flies using the CRISPRa system[34], which showed exactly similar eye phenotype as the *chm-Tip60-Gcn5* KD when driven by GMR-Gal4 (Fig. 6e). To further validate that the defective eye phenotype was indeed caused by elevated Wg signaling, we depleted *wg* in addition to the triple knocking down of *chm, Tip60* and *Gcn5*. Compared to the *chm-Tip60-Gcn5-GFP*-RNAi control, which was the same as the *chm-Tip60-Gcn5* KD phenotype, we observed that knocking down *wg* significantly rescued the eye phenotype caused by depletion of HATs. Similarly, simultaneous depletion of the three HATs and *arm*, encoding a key component of the Wg signaling pathway, dramatically rescued the defective eye phenotype caused by depletion of these HATs (Fig. 6e). Taken together, these analyses suggest that *chm, Tip60* and *Gcn5* synergistically play critical roles in regulating eye development, mainly through modulating the Wnt signaling pathway.

## Discussion
Armed with a variety of highly sophisticated genetic tools, *Drosophila melanogaster* is a powerful model organism for investigating the roles and regulation of genes in a broad spectrum of biological contexts. Loss-of-function and gain-of-function analyses provide the mechanistic insights into both the biological and the pathological processes. Compared to the gain-of-function approaches, loss-of-function analyses are less prone to potential artifacts, and many different techniques have been continuously developed to achieve this goal. A major breakthrough in recent years for loss-of-function analysis is the CRISPR-Cas9 genome editing system, which precisely generates mutation in the locus of interest using the endonuclease Cas9 guided by specific sgRNA. Despite the advantages of this technique, the mutant alleles generated by this method often produce strong biological consequences, such as lethality before adulthood or sterility if the gene product is required for gametogenesis. In addition, the CRISPR-Cas9 technique can have potential off-targets generated by mismatch between sgRNA and genome DNA, and bacterial Cas9 protein could be toxic, particularly in neurons. In fact, to be able to analyze a series of hypomorphic alleles or to achieve reduction of function, instead of complete loss of function of

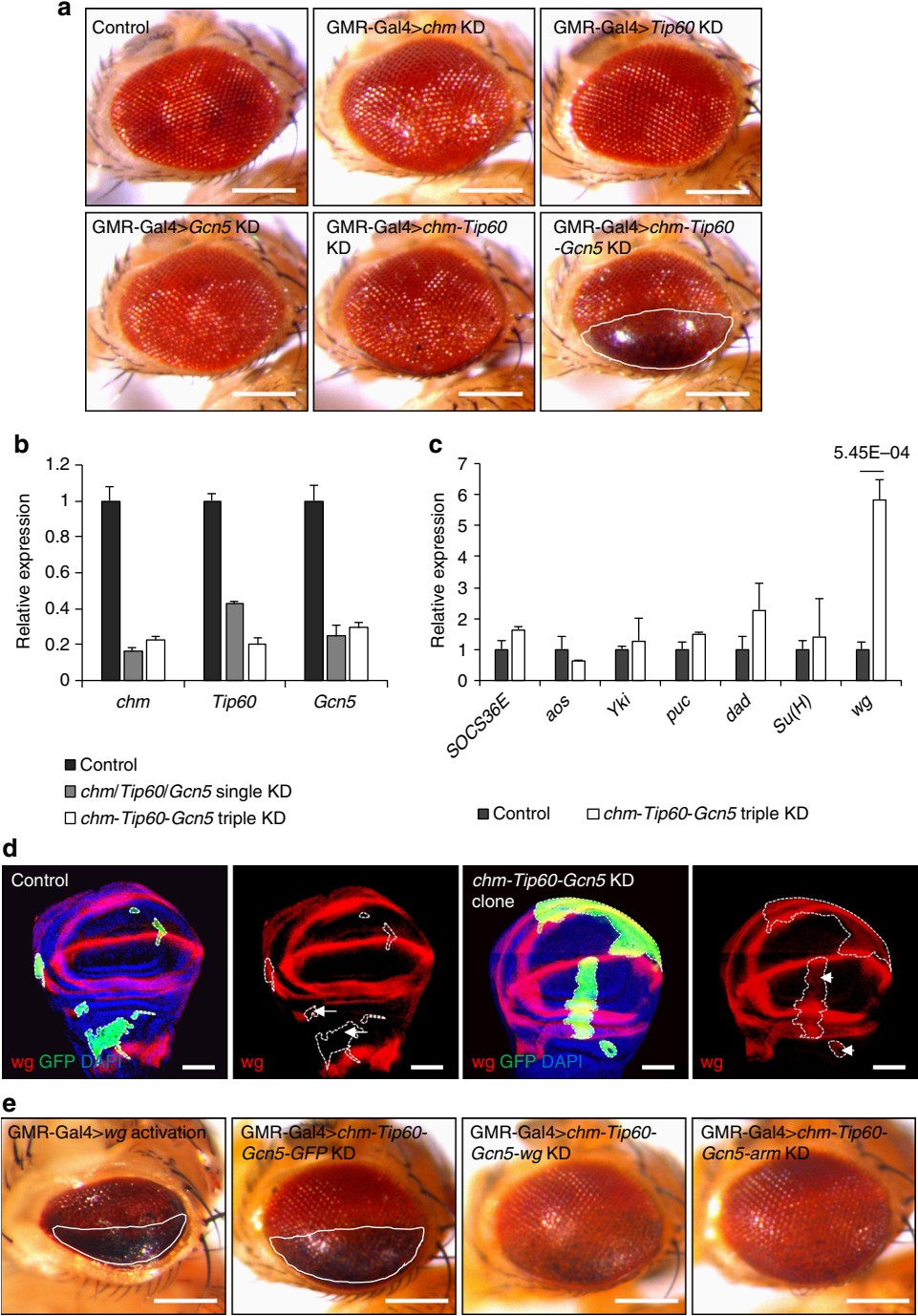

**Fig. 6** Histone acetyltransferases regulate eye development through Wnt signaling pathways. **a** *chm, Tip60, Gcn5* single knock down or *chm-Tip60* double knock down exhibited wild-type phenotype driven by GMR-Gal4, while simultaneous knock down of *chm-Tip60-Gcn5* using the pNP system produced severe eye defects. Scale bars, 200 μm. **b** qRT-PCR results show that *chm, Tip60, Gcn5* are all reduced to comparable level in single gene knockdown or triple genes knockdown flies. (*n* = 3, mean ± s.d.). **c** Wnt signaling pathway was significantly up-regulated as represented by Wg via qRT-PCR assay (*n* = 3, mean ± s.d.). Data are evaluated using one-tailed Student's *t*-test. **d** Immunostaining result showed up-regulated Wg in GFP-marked *chm-Tip60-Gcn5* KD clones. Dotted lines showed the GFP marked clone. Compared with control (indicated by long tail arrows), Wg signals were significantly increased in *chm-Tip60-Gcn5* KD clone (indicated by short tail arrows). Scale bars, 50 μm. **e** Transcriptional activation of *wg* produced similar eye phenotype as *chm-Tip60-Gcn5* KD flies. Knockdown of *wg* or *arm* at the background of *chm-Tip60-Gcn5* triple RNAi clearly rescued the eye phenotype. Scale bars, 200 μm

interested genes is often beneficial to advance our knowledge about the spectrum of functions and regulation of proteins in development. Given the simplicity of its design, high efficiency in knocking down genes of interest, and high penetration in phenotypic consequences, we regard the *Drosophila* transgenic RNAi as a powerful complement to the classical null alleles or strong

loss-of-function mutants generated by the CRISPR-Cas9 technique in deciphering the functions of genes. Transgenic RNAi technique is one of the most important genetic tools in *Drosophila* with the advantages of simplicity and high efficiency, and can knock down target genes at special developmental stage and particular tissues. In the past decade, several conditional

transgenic RNAi vectors have been developed, along with large-scale transgenic RNAi resources[1,4], which have greatly accelerated the progress of the *Drosophila* research community. However, there is still room for these existing transgenic RNAi systems to be further optimized. For example, it is puzzling to note that 15.8% of the RNAi lines generated so far can only be maintained as heterozygous (Fig. 2f). These problems have made genetic analysis difficult, especially when combining these RNAi lines with different Gal4 lines or overexpression lines. In this study, we identified the key issue of this problem is the leaky expression from the basal promoter, even without the Gal4 protein. To solve this challenge, we replaced the *hsp70* promoter with a DSCP promoter, thereby significantly reducing the rate of transgenic RNAi heterozygotes from 15% down to 0.6% (Fig. 2f).

Given that many gene products in *Drosophila* can have partially redundant functions in regulating different biological processes, thus depleting one type of these protein is often not effective to produce obvious phenotype. All the previous transgenic RNAi techniques can only carry a single shRNA; thus simultaneously targeting two or more genes requires multiple generations of genetic crosses to recombine them together, which can also result in unhealthy recombinants. In addition, genetic recombination may also reduce RNAi efficiency because of Gal4 dilution. Furthermore, combining three or more RNAi lines is not feasible in practice. The pNP vector developed in this work can carry multiple shRNAs and efficiently deplete different genes in parallel, thereby producing strong and additive phenotypes. Our case study of targeting three HATs illustrated the advantages of this transgenic RNAi system over the previous methods. Simultaneously depleting three HATs generated a severely defective eye phenotype, and our subsequent analyses revealed that hyperactive Wnt signaling caused by reduction of HATs accounts for these strong phenotypes (Fig. 6).

Although the transgenic RNAi technique is a powerful approach to studying gene function, caution is necessary to avoid potential false negative or false positive results. False negative results may be obtained particularly when performing genetic screens using the large collection of transgenic RNAi lines. One major factor that may contribute to this problem is the difficulty in depleting highly expressed genes to the threshold sufficient to cause phenotypic consequences. A key advantage of the pNP system, as shown in this work, is that it is much more efficient at knocking down highly expressed genes such as *aTub67C* than the previously developed pVALIUM20 system (Fig. 3c and Fig. 4). Furthermore, in the few cases that we tested, the pNP system yielded more clearly visible phenotypes than the pVALIUM20 system as one would expect (Fig. 3 and Fig. 4), suggesting that the RNAi from pNP system is more efficient and, therefore, the transgenic lines based on this system may help to reduce false negative rates.

Another question that needs caution is related to the false positive results. For example, some of the transgenic RNAi lines based on the pVALIUM20 system can cause lethality at different developmental stages, with or without combination with certain tissue-specific Gal4 lines[4,6,9,10]. Although this phenomenon is usually interpreted as the consequence of transgenic RNAi against a developmentally essential gene, our comparisons between the pVALIUM20 and pNP systems, together with the temperature-sensitive Gal80 repressor, suggest that leaky and non-specific expression from the basal promoter in the pVALIUM20 vector plus Gal4 induced RNAi in early developmental stage can cause lethality. Our analyses with the pNP transgenic lines suggest that the pNP system is tighter, and therefore more specific in tissue-specific knockdown than the pVALIUM20 system. Thus, we expect that using the transgenic RNAi lines based on the pNP system may help to reduce false positive rates.

Finally, inspired by the major advantages of the pNP system as characterized in this work, we have generated a collection of transgenic RNAi lines against the 1093 genes that are highly homologous to human disease genes, available from the Tsinghua Fly Center (Supplementary Data 4 and http://fly.redbux.cn/). We have performed a phenotypic analysis using MTD-Gal4, which induced RNAi depletion of these disease-associated genes specifically in germline, and then analyzed the effect on egg laying. We have analyzed 576 genes so far and 28.6% of the screened genes showed abnormal egg laying, suggesting the efficiency of this pNP system and the association of these genes with reproductive disease (Supplementary Data 5). We anticipate that this RNAi approach, as well as the relevant resources, will aid the *Drosophila* community to advance our understanding of the regulatory networks underpin on the normal developmental and physiological processes.

## Methods

**Vector and DNA constructs.** pVALIUM20 was cut with SpeI to remove the gypsy, Loxp, UAS and *hsp70* promoter, and then was ligated with an oligonucleotide fragment generated by annealing the two primers that contained EcoR I, Spe I, Bgl II, Avr II, Xba I, Nhe I sites (fwd: 5′-CTATGCTAGCAAACTAGTCAGATCTTG GAATTCATCTAGACCCTAGGAC-3′, rev: 5′-GTCCTAGGGTCTAGATGAATT CCAAGATCTGACTAGTTTGCTAGCATAG-3′); we named the vector V20-MCS1. The V20-MCS1 vector was cut with XhoI/KpnI and ligated with a small fragment cut from pEGFP-N1 with the same enzymes. The resulting vector was mutated at EcoRV, BamHI, XhoI, KpnI, SacI, MfeI sites by AccuPrime Pfx DNA Polymerase (Invitrogen) (fwd1: 5′-GTGACTCCTGAAGCTCAAGCTTCGAATTC TGCAGTCGACGGGTCCGTCGA-3′, rev1: 5′-TCGACGGACCCGTCGACTGCA GAATTCGAAGCTTGAGCTTCAGGAGTCAC-3′, fwd2: 5′-GATATTGAGTTCC AGACGAGCCACCAGTGCCCAACTGTTGGCGTCCAATCAT-3′, rev2: 5′-ATG ATTGGACGCCAACAGTTGGGCACTGGTGGCTCGTCTGGAACTCAATAT C-3′, fwd3: 5′-GCCATACCATTTAGCCGATCGCTTGTGCTCGGCAACAGTA T-3′, rev3: 5′-ATACTGTTGCCGAGCACAAGCGATCGGCTAAATGGTATGG C-3′). We named the intermediate vector V20-MCS2. V20-MCS2 was digested with NheI and ligated with a Loxp2 fragment (fwd: 5′-CTAGTATAACTTCGTA TAATGTATGCTATACGAAGTTATG-3′, rev: 5′-CATAACTTCGTATAGCATA CATTATACGAAGTTATACTAG-3′); the resulting vector was digested with NcoI/NheI to generate a small fragment and ligated with V20-MCS2 that was digested with the same enzymes. We named this intermediate vector V20-MCS2-Loxp2.

pVALIUM22 was cut with BamHI and ligated itself; then was cut with XbaI/ SacII and ligated with an amplified fragment from pVALIUM3 that contained U promoter and 5*UAS (fwd: 5′-AACCGCGGAGTACTGTCCTCCGAGCG-3′, rev: 5′-GGTCTAGACTTTGGTATGCGTCTTGTGAT-3′). The resulting vector was cut with SacII/PstI and ligated with another amplified fragment from pVALIUM20 that contained 5*UAS (fwd: 5′-TTCTGCAGGTCGGAGTACTGTCCTCC-3′, rev: 5′-AACCGCGGGAGTCTCCGCTCGGAG-3′); we named this intermediate vector V22-UAS-Up. pIB (Invitrogen) was cut with HindIII/XbaI and ligated with a small DNA fragment containing NheI, SacI, XbaI sites. The resulting vector was digested with NheI/SacI and ligated with a Loxp1 fragment (fwd: 5′-CTAGCATAACTT CGTATAATGTATGCTATACGAAGTTATGAGCT-3′, rev: 5′-CATAACTTCG TATAGCATACATTATACGAAGTTATG-3′), then the vector was cut with XbaI/ SacI and ligated with the fragment that was cut from V22-UAS-Up. We named the vector pIB-UAS-Up-Loxp1.

V20-MCS2-Loxp2 was digested with XbaI/NheI and ligated with a fragment, which was cut from pIB-UAS-Up-Loxp1 with XbaI/NheI. The resulting vector was digested with NheI and ligated with a gypsy fragment, which was amplified from V20-MCS2 (fwd: 5′-TTACTAGTCTGGCCACGTAATAAGTG-3′, rev: 5′- TTA CTAGTGTTGGTTGGCACACC-3′). We named the vector V20-Loxp1-gypsy-Loxp2-UAS-Up.

pVALIUM20 was digested with BamHI and ligated itself. The resulting vector was cut with SpeI and ligated with an oligonucleotide fragment generated by annealing the two primers that contained SpeI, AvrII et al. sites (SpeI beside AvrII) (fwd: 5′-CTAGTAGGGATCCACTCGAGAAGATCTAGGTACCAGCGGCCG CAT-3′, rev: 5′-ATGCGGCCGCTGGTACCTAGATCTTCTCGAGTGGATC CTACTAG-3′); we named it V20-BamHI-MCS1.

V20-Loxp1-gypsy-Loxp2-UAS-Up was digested with MfeI/XbaI and ligated with the shortest fragment cut by the same enzymes from V20-BamHI-MCS1; the resulting vector was cut with MfeI and ligated with the fragment digested from V20-BamHI-MCS1 with the same enzyme. We named this intermediate vector V20-MCS3. V20-MCS3 was digested with AvrII and ligated with a linker fragment which was amplified from the genome (fwd: 5′-AATCTAGACGCGATGCTC AAGGCAAAAA-3′, rev: 5′-TTCCTAGGCAGCATGAATGCGCCAATAT-3′). We named the final vector pNP (Supplementary Fig. 1 and Supplementary Note 1).

**Fly strains**. GMR-Gal4, ey-Gal4 were used to drive expression in the eye. Nub-GAL4, MS1096-Gal4, C96-Gal4, Nub-Gal4; tub-Gal80[ts] were used to drive expression in the wing. MTD-Gal4 and nos-Gal4 drove expression in the ovary and nos-Gal4 was also used in the testis. 1824-GAL4 (Bloomington #1824) drove expression in the salivary gland. esg-Gal4 and esg-Gal4, UAS-GFP; tub-Gal80[ts] was used to drive expression in the gut. elav-Gal4 was used to drive expression in the brain. act-Gal4; tub-Gal80[ts] was used to perform qRT-PCR assay of *histone H1, H2A, H2B, H3, H4* lines. y, hs-FLP; act5C < STOP < GAL4, UASp-GFP/CyO was used to perform *chm-Tip60-Gcn5* triple KD clone assays. Detailed genetic information is available at FlyBase (http://flybase.org/).

**Egg-laying assay**. To perform egg-laying assay, we collected virgin females within 6 h of eclosion, and mated ten virgin females with eight w1118 males per vial. Later, every female was transferred to new vials, ten vials were scored for each group. The flies were transferred to new vials containing fresh food every day. Then we counted the number of eggs laid by individual female from day three to nine, and one time for one day. Approximately 10 vials were scored for each group.

**qPCR assay**. RNA that from salivary glands, wing discs, eye discs of L3 larvae, and ovary or entire flies in adult stage were isolated using TRIzol (Invitrogen). The total RNA was treated with DNaseI and 1 μg RNA subjected to reverse transcription using the GoldScript cDNA Assay Kit (Invitrogen). qPCR reaction was performed by using SYBR Premix Ex Taq (TAKARA), and the results were analyzed with the iQ5 real-time PCR detection system (Bio-Rad). Calculation of steady state RNA levels was by the 2-ΔΔCt method. The primers used for qPCR are listed in Supplementary Data 6.

**Luciferase assay**. Luciferase activity was measured using the Steady-Glo Luciferase Assay Kit (Promega). 40 embryos, five wandering L3 larvae, five pupae or five female flies (2-day-old) were collected in 300 μL GLO lysis buffer for each sample; each luciferase assay contained five independent samples. Muscle and fat body of five L3 larvae, 20 brain discs, 40 wing discs, 40 gonads, 20 salivary glands of L3 larvae, and 10 guts from female flies were dissected in cold PBS and transferred to 90 μL GLO lysis buffer for each sample; three independent samples were used for each assay. Samples homogenized and then centrifuged at 12,000×g for 15 min, and then take 50 μL supernatant in 96-well plates. After incubation in the dark for 20 min, luminescence was measured on a luminometer (Thermo Scientific, VARIOSKAN FLASH).

**Polytene chromosome immunostaining**. The third-instar larval polytene chromosomes was dissected and immunostained[36]. Chromosomes were incubated with mouse monoclonal anti-HP1a (HP1a C1A9, 1:100) or rabbit polyclonal anti-acetyl-histone H4 Antibody (EMD Millipore, 06-866, 1:200). After incubation for 1 h at room temperature, chromosomes were washed three times in PBS and incubated with a goat anti-mouse antibody conjugated to FITC (DSHB, 1:100) and a goat anti-rabbit antibody conjugated to FITC (DSHB, 1:100) for 2 h (room temperature). After one PBS wash, the slides were incubated with 4,6-diamidino-2-phenylindole (Sigma, 1: 1000) to stain DNA. The slides were mounted in Fluoromount mounting media (Sigma, F4680), and the immunostaining images were performed by using a Nikon ECLIPSE Ti inverted microscope (×100 model). Adobe Photoshop software was used for measurements.

**Wing disc immunostaining**. Clones of RNAi cells in the wing disc were generated by FLP/FRT-mediated mitotic recombination[37]. To generate clones, y, hs-FLP; act5C < STOP < GAL4, UASp-GFP/CyO was crossed with the UAS-Gcn5-Tip60-chm triple RNAi line at 25 ℃. L1 stage larvae were heat-shocked at 37 ℃ for 1 h. Then the larvae were cultured at 29 ℃ until the L3 stage. The L3 stage larvae were dissected in PBS for immunostaining. Wing discs were incubated with rabbit monoclonal anti-GFP (Abcam, 1:1500) and mouse polyclonal anti-acetylated wingless (DSHB, 1:500) at 4 ℃ overnight. Then the samples were washed five times in PBT for 15 min each, and incubated with goat anti-mouse antibody conjugated to TRITC (DSHB, 1:100) and a goat anti-rabbit antibody conjugated to FITC (DSHB, 1:100) for 2 h at 25 ℃. The samples were incubated with 4,6-diamidino-2-phenylindole (Sigma, 1: 1000) to stain DNA, then washed with PBT five times for 15 min each. The stained wing discs were mounted, and images were obtained with an inverted Zeiss LSM780 fitted with an ultraviolet laser. The images were processed by using the ZEN program.

**Stem cell immunostaining**. Immunostaining of ovaries, testis and intestine was performed following standard protocols[36,38,39]. The treatment and control group were cultured at 29 ℃ for 10 days. Then dissected them in cold PBS for immunostaining. The following primary antibodies were used: mouse monoclonal anti-Hts antibody 1B1 (Developmental Studies Hybridoma Bank (DSHB), 1:100), rabbit polyclonal anti-Vasa (Santa Cruz Biotechnology, sc30210, 1:200), Mouse monoclonal anti-Pros (DSHB, 1:200), rabbit monoclonal anti-GFP (Abcam, 1:1500). Images were obtained with Zeiss LSM780 fitted with an ultraviolet laser.

**Construction of transgenic RNAi plasmids**. Detailed information on primers can be found in Supplementary Data 7. For the large scale RNAi constructs, primers were ordered in a 96-well plate format and annealed at 95 ℃.The pNP vector was digested with EcoRI/NheI, and ligated with the annealed primers. Using PCR to select clones (fwd: 5′-GCTGAGAGCATCAGTTGTGA-3′, rev: 5′-TAATCGTGT GTGATGCCTACC-3′), the correct clones were PCR for a 750 bp band and the wrong clones showed a 1000 bp band. Correct constructs were confirmed by DNA sequencing; the primer was: 5′-GCTGAGAGCATCAGTTGTGA-3′.

For the more than one short hairpin constructs, we firstly cloned all short hairpins into the pNP vector as mentioned above; one resulting vector was linearized by digesting with SpeI and another vector was digested with SpeI and XbaI to get the fragment (300 bp) containing a short hairpin. The resulting products were purified (AxyPrep DNA Gel Extraction Kit) and ligated each other. Following transformation, correct clones were selected by PCR (fwd: 5′-GCTGAG AGCATCAGTTGTGA-3′, rev: 5′-TAATCGTGTGTGATGCCTACC-3′) and further confirmed by restriction enzyme digestion (SpeI and XbaI). The resulting vector could be cut with SpeI and ligated with another short hairpin fragment which was cut with SpeI and XbaI from another vector.

**Construction of transgenic overexpression plasmids**. To generate RNAi insensitive overexpression lines, we cloned the vector by using Gibson assembly method. First we cut the pVALIUM20 and pNP vector by EcoRI/NheI enzymes, PCR the coding sequences of E2F1, Upf1, aTub67C, egg, and chm-T2A-Tip60-T2A-Gcn5, then we used the Hieff Clone Plus Multi One Step Cloning Kit (Yeasen, CAT:10912ES10) to assemble them together, finally we performed site directed mutagenesis by PCR. PCR primers for all can be found in Supplementary Data 8.

**Construction of *wg* and *chm-Tip60-Gcn5* activation vector**. *wg* and *chm-Tip60-Gcn5* activation were achieved by using the CRISPRa system, and the construct was generated as described in Jia et al. (2018)[34]. Primer sequences for sgRNAs are: *wg*-fwd: 5′-TTCGCCGACGATGCGATCGGATCG-3′, *wg*-rev: 5′-AAACCGATCCGATCGC ATCGTCGG-3′; *chm*-fwd: 5′-ttcgATCGTACTCGACAAGCACTC-3′, rev: 5′-aaacGA GTGCTTGTCGAGTACGAT-3′; *Tip60*-fwd: 5′-ttcgGCAGTGTTGCTAACTACGC T-3′, rev: 5′-aaacAGCGTAGTTAGCAACACTGC-3′; *Gcn5*-fwd: 5′-ttcgTTGACTAC GCCCCTGCTAAT-3′, rev: 5′-aaacATTAGCAGGGGCGTAGTCAA-3′.

**Statistics analysis**. GraphPad Prism and Excel was used to calculate the mean and the s.d. for all the column diagram data, as well as the statistical analysis. For the *n* of each test, and detailed statistical analysis method is showed in the figure legend.

## Data availability
The data that support the findings of this study are available from the corresponding author upon reasonable request.

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

## Acknowledgements

This work was supported by the National Key Technology Research and Development Program of the Ministry of Science and Technology of the People's Republic of China (2016YFE0113700), and the National Natural Science Foundation of China (31571320, 91729301, 81630103), and the National Science Fund for Distinguished Young Scholars (31725023).

## Author contributions

J.-Q.N. designed the experiment. H.-H.Q., F.W., R.-G.X., J.S., R.Z., X.R., D.M., X.W., Y.J., P.P., D.S. and L.-P.L. performed the experiments. H.-H.Q., R.-G.X., F.W., J.-Y.J., Q.L. Z.C., G.W. and S.L. performed data analysis. J.-Q.N. wrote and revised the manuscript.

## Additional information

**Competing interests:** The authors declare no competing interests.

