## [Peer Review File · Nature Communications]

Reviewers' Comments:

Reviewer #1:

Remarks to the Author:

In this manuscript, Qiao et al. developed and characterized a transgenic RNAi system based on a novel pNP vector. This new system is developed because previous RNAi systems had false positive and negative results with relatively high toxicity. Their findings suggest that this novel system works both in somatic and germline systems with high efficacy. With this system, one can delete multiple genes while allows to study the functions of redundant genes or protein complexes. In addition, using this system, they focused on all histone acetyltransferases (HATs) together because it produced no eye phenotype when knocking down an individual HATs. Their data suggest that HATs work together in regulating Wnt signaling. Finally, they also generated a resource for genes that has close function to human disease genes. In general, this provide a novel RNAi system to eliminates the false positive or false negatively as compared to previously generated system. However, following modifications are needed before consideration.

1. Introduction paragraph 1: "several large Drosophila transgenic RNAi collections have been generated, and their applicationsgenes in developmental and physiological". Authors have NOT provided any references.
2. Similarly, authors have not provided any references in following sentence in Introduction: "These applications by many fly laboratories simultaneously target multiple genes".
3. The abbreviated name of genes should be defined at first use.
4. Results: Authors noticed "15.8% of RNAi lines using pVALIUM20 system in the Tsinghua Fly center (THFC) cannot maintain homozygosity. 8.5% of the ϕ C31 and dsRNA RNAi library (KK collection) VDRC are heterozygous, and 17.9% of the pVALIUM20 system based RNAi lines in the VDRC are heterozygous". Authors should provide these lines information in a supplementary file.
5. Results: Each Supplementary Table should have a title to explain, what is presented in the table.
6. Results: Authors wrote "we constructed more than 300 pNP based transgenic lines, targeting the same genes with the same shRNAs as the pVALIUM20 heterozygous rate of these transgenic lines". Authors should provide these 300 pNP based transgenic lines in a supplementary file.
7. In Supplementary Figure 4, level of H4ac has been shown to be reduced in the triple gene knock down of chm-Tip60-Gcn5, however, based on DNA staining it appears that number of cells are also reduced.
8. Authors have generated a collection of transgenic RNAi lines against the 1093 genes that are highly homologous to human disease genes, however, in a supplementary file provided by authors have not shown what human diseases these genes are associated. It should be provided.
9. RNAi knockdown studies at the genome scale have been demonstrated in the neurons, intestine, testis and ovarian stem cell system. Authors have not tested any of the genes identified using this new RNAi system and not even referenced.
10. Authors need to provide references in the contest of presented in the introduction and discussion section. For e.g. some of the transgenic RNAi lines based on the pVALIUM20 system can cause lethality...
11. Authors have not provided any phenotypic data on the lines generated for human disease genes in the supplementary file.

Reviewer #2:

Remarks to the Author:

In this manuscript, Qiao et al. develop and test a new vector for transgenic expression of RNAi constructs. The new vector, named pNP contains a shorter basal promoter than the current state of the art RNAi vector pVALIUM20. This results in lower leaky expression, that the authors claim is an important problem with pVALIUM20. Additionally, the new vector allows expression of two RNAi

constructs from the same transgene, a feature lacking from previous RNAi vectors. They show that knocking down multiple targets with this vector results in a more reliable knock down than using multiple individual transgenes. They identify Histone Deacetylases play a redundant role in regulation of Wnt signaling in eye development as a proof of concept. Finally the authors generate a sizeable collection of 1000 transgenic stocks targeting genes implied in human disease. All in all, the construct is novel and the improvements to the current state of the art and the collection is useful for the fly community. If the issues listed below can be addressed I am in favor of publishing this work.

Major points are:

- The heterozygosity of the lines in stock centers are used as an argument for leaky expression of hairpins under the control of pVALIUM20. This suggestion is unsubstantiated. Second site mutations in chromosomes are rife (up to 50% of the EMS induced mutations share a second site lethal). The pVALIUM20 constructs can be the cause of the lack of homozygosity but second site lethal mutations probably cause the majority of lethal phenotypes. First, these chromosomes often share the same lethals because the stocks were not isogenized prior to injection (see Bellen et al., 1989). Simple complementation tests will reveal if this is the case or not. Alternatively they should test whether the lethality can be rescued by co-expressing the target gene from *pseudobscura* (Langer, Ejsmont and Schonbauer et al. 2010). The authors indicate that they cloned 300 pNP RNAi constructs targeted by pVALIUM and made transgenics to test whether they get homozygous. It is not clear whether they have generated independently the same 300 constructs in pVALIUM, inserted in the same locus and compared the lethality or they compare the lethality to the available transgenic stocks in stock centers. If they generated independently the 300 stocks in pVALIUM, the difference maybe the stocks used. Did they isogenize prior to injection?
- The authors compare the leakiness of the pVALIUM and pNP by placing luciferase reporter under the control of promoters/enhancers used in these vectors. The luciferase levels in animals where the expression is induced by a GAL4 should be measured and compared too for 5 drivers.
- There is no comparison of RNAi knock down phenotypes to know null mutants of the targeted genes. The authors take a stronger phenotype as the correct phenotype associated with the gene which is not trivial in all cases. Stronger phenotypes can be the result of higher expression levels of RNAi constructs targeting non-specific genes, or additive non-specific targeting from multiple RNAi constructs on the same transgene. Ideally the authors should rescue some of the phenotypes by expressing corresponding fosmids from *Drosophila pseudoobscura* fosmid library (Langer, Ejsmont and Schonbauer et al. 2010) for a subset of targets. This is important for the HDAC experiment where the authors use triple knock-down construct.
- Phenotypes depicted in Figure 3 a-d need to be quantified and rescued.
- The argument about H2A, 2B and 4 by pVALIUM being lethal due to leaky expression is obviously confusing. Leaky expression of Hs70 basal promoter does not require GAL4 activity and would not be responsive to the GAL80ts that the authors use to silence expression during development. Moreover, a later shift in temperature only causes mild phenotypes. On the one hand the authors are suggesting that the basal expression level is enough to cause lethality (see above) on the other the induced expression levels are not strong enough to cause a phenotype. These are inherently conflicting arguments. A more likely explanation is the genetic background as stated above.
- In the discussion the authors indicate low knock down levels rates by somatic expression of gRNAs. This is misleading. Port et al. showed efficient editing in somatic tissues using transgenic sources of gRNAs. RNAi and somatic gRNAs can be used as complementary approaches. There is no need to mislead the readers into thinking that one works and the other does not. Please

remove: "387 Furthermore....screening".

In summary the technique and the resources generated in this manuscript are useful but the authors should substantiate some of their claims with some additional experiments.

Reviewer #1:

1. Introduction paragraph 1: “several large *Drosophila* transgenic RNAi collections have been generated, and their applicationsgenes in developmental and physiological”. Authors have NOT provided any references.

Response: Thank you for pointing out this. We have added related references following “several large *Drosophila* transgenic RNAi collections have been generated, and their applicationsgenes in developmental and physiological”.

2. Similarly, authors have not provided any references in following sentence in Introduction: “These applications by many fly laboratories simultaneously target multiple genes”.

Response: Yes, we realized this. We have added related references following this sentence in the main manuscript as you suggested.

3. The abbreviated name of genes should be defined at first use.

Response: Thanks for your comments, we have checked the full manuscript and defined each of abbreviated name in detail.

4. Results: Authors noticed “15.8% of RNAi lines using pVALIUM20 system in the Tsinghua Fly center (THFC) cannot maintain homozygosity. 8.5% of the ϕ C31 and dsRNA RNAi library (KK collection) VDRC are heterozygous, and 17.9% of the pVALIUM20 system based RNAi lines in the VDRC are heterozygous”. Authors should provide these lines information in a supplementary file.

Response: This is a good suggestion. The lines information used here have been provided in Supplementary Table 1.

5. Results: Each Supplementary Table should have a title to explain, what is presented in the table.

Response: We agree with this suggestion, we have provided titles to all Supplementary Tables.

6. Results: Authors wrote “we constructed more than 300 pNP based transgenic lines, targeting the same genes with the same shRNAs as the pVALIUM20 heterozygous rate of these transgenic lines”. Authors should provide these 300 pNP based

transgenic lines in a supplementary file.

Response: We have provided the information of these lines for both pVALIUM20 and pNP in Supplementary Table 2.

7. In Supplementary Figure 4, level of H4ac has been shown to be reduced in the triple gene knock down of *chm-Tip60-Gcn5*, however, based on DNA staining it appears that number of cells are also reduced.

Response: Thank you for this careful observation. Based on our immunostaining results in Supplementary Fig. 8 (Supplementary Fig. 4 in previous version), triple knockdown of *chm-Tip60-Gcn5* not only reduced the level of H4ac, but also generated much thinner chromosomes, indicating potential effects on DNA endoreplication. Meanwhile, the *chm-Tip60-Gcn5* triple knockdown salivary glands were indeed much smaller than controls when dissecting. We agree with reviewer that the small salivary glands reduced the cell number and/or DNA endoreplication, exploring the underlying mechanism of these effects is beyond the scope of this work, but it will be very interesting to further analyze this phenomenon in the future.

8. Authors have generated a collection of transgenic RNAi lines against the 1093 genes that are highly homologous to human disease genes, however, in a supplementary file provided by authors have not shown what human diseases these genes are associated. It should be provided.

Response: Thanks for your suggestion, we have included the human diseases information that these genes are associated with in a new column (Supplementary Table 4).

9. RNAi knockdown studies at the genome scale have been demonstrated in the neurons, intestine, testis and ovarian stem cell system. Authors have not tested any of the genes identified using this new RNAi system and not even referenced.

Response: We have cited the relevant references in the revised manuscript. Based on previous genome-scale RNAi screens in the neurons, intestine, testis and ovarian stem cell systems, we chose a set of protein-coding genes with known loss-of-function phenotypes to detect the efficiency of this new pNP RNAi system. As shown in

Supplementary Fig. 2a, knocking down of target genes in the neurons through *elav-Gal4* produced lethality phenotype, which is consistent with previous report (Dietzl, G. *et al.*, *nature*, 2007; Koizumi, K. *et al.*, *PNAS*, 2007; Parrish, J. Z. *et al.*, *Genes & development*, 2006). When we reduced the protein level of *notch* using *esg-Gal4* in the intestine, both intestinal stem cell and enteroendocrine cell showed hyperproliferation (Supplementary Fig. 2b), which is a typical Notch signaling defect phenotype (Zeng, X. *et al.*, *Cell reports*, 2015). In addition, RNAi of *punt* in the testis generated germ cells loss phenotype (Liu, Y. *et al.*, *Nature communications*, 2016), while *bam*-depleted ovary increased the number of spectrosome-containing cells (a tumor like phenotype) (Yan, D. *et al.*, *Developmental cell*, 2014) (Supplementary Fig. 2c, d). Altogether, these phenotypic assays support the novel pNP system works efficiently in the neurons, intestine, testis and ovarian stem cell system, which is consistent with previous RNAi screens.

10. Authors need to provide references in the context of presented in the introduction and discussion section. For e.g. some of the transgenic RNAi lines based on the pVALIUM20 system can cause lethality...

Response: Thanks for your kind remind, and we have added related references in these sections in the revised manuscript.

11. Authors have not provided any phenotypic data on the lines generated for human disease genes in the supplementary file.

Response: This is a very suggestive comment. We were carrying out a phenotypic analysis using MTD-Gal4 which induced RNAi of these human disease genes in female germline system, and tested the effect on egg laying. This is an ongoing effort and we have analyzed 576 genes so far and 28.6% of the screened genes showed abnormal egg laying, these results support the efficiency of this pNP system and the association of these genes with reproductive disease. Detail results are now included in Supplementary Table 5.

Reviewer #2:

1. The heterozygosity of the lines in stock centers are used as an argument for leaky expression of hairpins under the control of pVALUIM20. This suggestion is unsubstantiated. Second site mutations in chromosomes are rife (up to 50% of the EMS induced mutations share a second site lethal). The pVALIUM20 constructs can be the cause of the lack of homozygosity but second site lethal mutations probably cause the majority of lethal phenotypes. First, these chromosomes often share the same lethals because the stocks were not isogenized prior to injection (see Bellen et al., 1989). Simple complementation tests will reveal if this is the case or not. Alternatively, they should test whether the lethality can be rescued by co-expressing the target gene from *pseudobscura* (Langer, Ejsmont and Schonbauer et al. 2010). The authors indicate that they cloned 300 pNP RNAi constructs targeted by pVALIUM and made transgenics to test whether they get homozygous. It is not clear whether they have generated independently the same 300 constructs in pVALIUM, inserted in the same locus and compared the lethality or they compare the lethality to the available transgenic stocks in stock centers. If they generated independently the 300 stocks in pVALIUM, the difference maybe the stocks used. Did they isogenize prior to injection?

Response: Thanks for your comments and insightful suggestion. We agree with that second site lethal mutations could also result in the lethal phenotypes. To exclude the effect of second site mutations and test if the leaky expression of hairpins in pVALUIM20 is also the cause for the heterozygosity of the transgenic lines, we have performed the rescue experiment in the heterozygous transgenic RNAi flies through co-expressing the target genes that insensitive to the hairpins, and regulated by their endogenous promoters, as we are limited by the resources related to *Drosophila pseudoobscura* fosmids at the moment. For the four tested heterozygous lines, all of them can be rescued and generate homozygosity, suggesting the leaky expression of hairpins was also the cause for heterozygosity.

In our experiment to compare the homozygous rate, we have constructed RNAi lines using pNP and pVALIUM20 system in parallel to against more than 300 genes, and did the isogenize before injection. For each target gene, same shRNA was cloned

into pVALIUM20 or pNP vector respectively, and they were inserted in the same chromosome locus of lines, TB16 (*y, v, nanos-integrase; attP40*) or TB18 (*y, v, nanos-integrase; ; attP2*). We observed that the pVALIUM20 vector had a higher heterozygous rate than the pNP vector, and detailed information was listed in Supplementary Table 2. We thank the reviewer for pointing out this and we have revised the manuscript to clarify this point.

2. The authors compare the leakiness of the pVALIUM and pNP by placing luciferase reporter under the control of promoters/enhancers used in these vectors. The luciferase levels in animals where the expression is induced by a GAL4 should be measured and compared too for 5 drivers.

Response: This is a very helpful suggestion for our work. Accordingly, we have tested the expression of luciferase in different tissues of these transgenic flies using different Gal4 drivers, including *elav-Gal4* (nervous system), *MS1096-Gal4* and *Nub-Gal4* (wing), *MTD-Gal4* and *nos-Gal4* (germline), *GMR-Gal4* (eye) and *esg-Gal4* (gut). As shown in Supplementary Fig. 3, both *pNP-luciferase* and *pVALIUM20-luciferase* transgenic flies showed highly expressed luciferase comparing with control. In addition, the luciferase intensities from the *pNP-luciferase* transgenic flies are all significantly higher than *pVALIUM20-luciferase* animals using the corresponding drivers, suggesting that the novel pNP system is more efficient than the pVALIUM20 system, which is consistent with the phenotypic assay in Fig. 3.

3. There is no comparison of RNAi knock down phenotypes to know null mutants of the targeted genes. The authors take a stronger phenotype as the correct phenotype associated with the gene which is not trivial in all cases. Stronger phenotypes can be the result of higher expression levels of RNAi constructs targeting non-specific genes, or additive non-specific targeting from multiple RNAi constructs on the same transgene. Ideally the authors should rescue some of the phenotypes by expressing corresponding fosmid from *Drosophila pseudoobscura* fosmid library (Langer, Ejsmont and Schonbauer et al. 2010) for a subset of targets. This is important for the HDAC experiment where the authors use triple knock-down construct.

Response: This is indeed a very helpful suggestion. We agree with the reviewer that there is a possibility that the stronger phenotypes can be the result of RNAi targeting

non-specific genes, or additive non-specific targeting from multiple RNAi constructs. We are limited by the resources related to *Drosophila pseudoobscura* fosmids at the moment. To exclude this possibility and rescue the phenotype in the HDAC experiment, we overexpressed *chm-T2A-Tip60-T2A-Gcn5* simultaneously that insensitive to the hairpins by using the T2A self-cleaving peptide in the *chm-Tip60-Gcn5* triple KD flies, as we expected, the KD phenotype was fully rescued (Supplementary Fig. 7b), supporting the specificity of this RNAi experiment. Furthermore, we also generated a transgenic activation line that targeting these three genes simultaneously using the flySAM system we developed recently (Jia Y. *et al.*, *PNAS*, 2018). As shown in Supplementary Fig. 7c, activation of these three genes also significantly rescued *chm-Tip60-Gcn5* triple KD tumor-like phenotype, further supporting the specificity of the triple KD result. In addition, we are very interested in testing the idea of using *Drosophila pseudoobscura* fosmids as suggested by the reviewer in the future.

4. Phenotypes depicted in Figure 3 a-d need to be quantified and rescued.

Response: We appreciate very much for this suggestion. For each phenotype depicted in Fig. 3 a-b, more than 100 flies were analyzed and at least 80% of these flies produced the same phenotype as we showed in Fig. 3 a-b. Meanwhile, for each phenotype shown in Fig. 3 c-d, all the flies showed the same defect. Moreover, to exclude the possibility of off-target effect and rescue the phenotype generated in Fig. 3 a-d, we have overexpressed the target genes that are insensitive to the hairpins in the RNAi flies driven by specific Gal4 lines. As shown in Supplementary Fig. 4, both the pVALIUM20-induced phenotype and the pNP-induced phenotype were significantly rescued, compared to the RNAi phenotype alone, suggesting the specificity of these RNAi. These results support the efficiency and specificity of this novel pNP system.

5. The argument about H2A, 2B and 4 by pVALIUM being lethal due to leaky expression is obviously confusing. Leaky expression of Hs70 basal promoter does not require GAL4 activity and would not be responsive to the GAL80ts that the authors use to silence expression during development. Moreover, a later shift in temperature only causes mild phenotypes. On the one hand the authors are suggesting that the basal expression level is enough to cause lethality (see above) on the other the

induced expression levels are not strong enough to cause a phenotype. These are inherently conflicting arguments. A more likely explanation is the genetic background as stated above.

Response: Thanks for your comments. Some transgenic RNAi lines using pVALIUM20 system were heterozygous because of high basal expression and thus caused lethality when homozygous, but for pVALIUM20-*H2A*, *H2B* and *H4* used here, they can be generated in homozygous, indicating leaky expression of pVALIUM20-*H2A*, *H2B* and *H4* alone cannot cause lethality. We agree with reviewer that leaky expression of *hsp70* basal promoter does not require GAL4 activity and would not be responsive to the GAL80^{ts}. In our opinion, the lethality of pVALIUM20-*H2A*, *H2B* and *H4* flies is likely resulted from the combination of the leaky expression and MS1096-Gal4 induced RNAi from pVALIUM20 system in early developmental stage. In our analysis, pVALIUM20-*H2A*, *H2B* and *H4* driven by MS1096-Gal4 were all lethal (Supplementary Fig. 5), but were viable and generated weak wing phenotypes using Nub-Gal4; tub-Gal80^{ts}, which prevented RNAi in early developmental stage (Fig. 4a). However, these hairpins driven by MS1096-Gal4 in the more efficient pNP system (also with low leaky expression) were viable (Supplementary Fig. 5) and gave the same severe phenotypes as Nub-Gal4; tub-Gal80^{ts} generated (Fig. 4a). Taken together, these results suggest that the leaky expression of the pVALIUM20 system plus MS1096-Gal4 induced RNAi in early developmental stage can cause lethality. Nevertheless, we have revised the manuscript to further clarify this point.

6. In the discussion the authors indicate low knock down levels rates by somatic expression of gRNAs. This is misleading. Port et al. showed efficient editing in somatic tissues using transgenic sources of gRNAs. RNAi and somatic gRNAs can be used as complementary approaches. There is no need to mislead the readers into thinking that one works and the other does not. Please remove: “387 Furthermore....screening”.

Response: Thanks for your kind remind. To avoid the mislead to the readers we have removed this sentence accordingly.